# Prenatal Diagnosis of Placental Mesenchymal Dysplasia with 46, X, Isochromosome Xq/45, X Mosaicism

**DOI:** 10.3390/genes13020245

**Published:** 2022-01-27

**Authors:** Chin-Chieh Hsu, Chien-Hong Lee, Shuenn-Dyh Chang, Tsang-Ming Ko, Shir-Hwa Ueng, Yu-Hsiu Chen, Mei-Chia Wang, Yao-Lung Chang

**Affiliations:** 1Department of Obstetrics and Gynecology, Chang Gung Memorial Hospital, Linkou Medical Center, College of Medicine, Chang Gung University, Taoyuan 33302, Taiwan; b9602047@cgmh.org.tw (C.-C.H.); gene@cgmh.org.tw (S.-D.C.); 2Department of Laboratory Medicine, Chang Gung Memorial Hospital, Linkou Medical Center, College of Medicine, Chang Gung University, Taoyuan 33302, Taiwan; li5592@cgmh.org.tw (C.-H.L.); ottermika@gmail.com (M.-C.W.); 3Genephile Bioscience Laboratory, Ko’s Obstetrics and Gynecology, Taipei 10050, Taiwan; tsangming@live.com; 4Department of Pathology, Chang Gung Memorial Hospital, Linkou Medical Center, College of Medicine, Chang Gung University, Taoyuan 33302, Taiwan; shu922@cgmh.org.tw (S.-H.U.); cyh0914@cgmh.org.tw (Y.-H.C.)

**Keywords:** placental mesenchymal dysplasia, biparental/androgenetic mosaicism, isochromosome X, 45 X, array CGH, microsatellite, short tandem repeats, karyotype

## Abstract

Placental mesenchymal dysplasia is an uncommon vascular anomaly of the placenta with characteristics of placentomegaly and multicystic appearance and with or without association with fetal chromosomal anomaly. We present a unique placental mesenchymal dysplasia patient with amniotic fluid karyotyping as 46, X, iso(X) (q10). Detailed molecular testing of the amniotic fluid, fetal cord blood, non-dysplastic placenta and dysplastic placenta was conducted after termination of pregnancy, from which we proved biparental/androgenetic (46, X, i(X) (q10)/45, X) mosaicism in different gestational tissues. A high portion of androgenetic cells in dysplastic placenta (74.2%) and near 100% of biparental cells in the fetus’s blood and amniotic fluid were revealed. Delicate mosaic analyses were performed, and possible pathogenesis and embryogenesis of this case were drawn up.

## 1. Introduction

Placental mesenchymal dysplasia (PMD), also known as “pseudo hydatidiform mole”, is characterized by a large placenta with grape-like, multicystic change [1]. This rare disease is often misdiagnosed as a partial molar pregnancy due to similar appearance under ultrasonography. However, the karyotype of a partial mole is usually triploid with multiple anomalies of the fetus. In contrast, PMD can show a diploid karyotype of fetus and structurally normal fetus. Distinguishing between the two etiologies mentioned above is important because a normal fetus could be born alive from a pregnancy with PMD, though 33% of them will have intra-uterine growth restriction, and 52% will have preterm birth. Peripartum complications also occur in mothers who have PMD, including gestational hypertension and pre-eclampsia [2].

Abnormal genetic composition of the fetus is seen in one-third of cases with PMD, including aneuploidy, uniparental disomy and Beckwith–Wiedemann syndrome. Additionally, by using molecular techniques, biparental/androgenetic mosaicism is ascertained in dysplastic placenta in some literature, which results from aberrant embryogenesis of a single zygote [3]. Here, we present a patient who was diagnosed as PMD and was carrying an euploid female fetus with a chromosomal anomaly detected by amniocentesis as 46, X, i(X) (q10). The dysplastic placenta showed high level of androgenetic 45, X mosaicism. Serial molecular analyses, including cytogenetic studies, array comparative genomic hybridization (array CGH) and short tandem repeat (STR) analysis were undertaken in order to demonstrate gradient change of the two mosaic cell lines and the possible route of pathogenesis.

## 2. Materials and Methods

### 2.1. Patient

Here, we present a 35-year-old female, who was pregnant at 15 weeks of gestation when she first visited our clinic. This is her second pregnancy, and her previous pregnancy ended with spontaneous abortion at early gestation. She was pregnant after using clomiphene. At 15 weeks of gestation, ultrasonography showed multicystic changes in the placenta with a viable, normally structured fetus (Figure 1a). The placental cysts ranged in size from 0.5–1.0 cm. Adequate fetal growth and normal amniotic fluid amount were seen. Biochemistry study from maternal serum revealed β-hCG as 58,247 mIU/mL and an elevated α-fetoprotein of 149.3 ng/mL (which was 3.73 multiple of median (MoM)) [4]. Based on ultrasound findings and an elevated maternal serum α-fetoprotein, PMD was suspected, and detailed prenatal genetic testing was suggested. Amniocentesis was performed at 17 weeks, which revealed the karyotype as 46, X, isochromosome Xq10. Array CGH confirmed the copy number variation as Xp deletion and Xq duplication. Level 2 ultrasound showed normal fetal structure and an enlarged placenta containing two parts: dysplastic and non-dysplastic parts (Figure 1b). Low vascularity was detected in the dysplastic placenta via Doppler scan and 3D reconstruction (Figure 1c,d). Genetic counseling was given, and the mother decided to terminate the pregnancy due to an isochromosome of the long arm of chromosome X, a Turner syndrome variant. An abortus with its dysmorphic placenta was delivered smoothly. The fetus’s general appearance was normal without omphalocele, macroglossia or appearing macrosomic.

The gestational tissue, including the amniotic fluid from prenatal amniocentesis, cord blood and placental biopsy after delivery, were collected and sent for further cytogenetic and molecular evaluation, and the remaining placenta was fixed for histopathological examination.

### 2.2. Cytogenetic Studies: Cell Culture and Karyotyping

For amniotic fluid cells and placenta tissue, long-term culture (>7 days) was applied with Chang medium T101-019 (Irvine Scientific, Santa Ana, CA, USA). For cord blood, Phytohemagglutinin Assay (PHA) was used to stimulate mitosis of peripheral lymphocytes under short-term culture (2–3 days) with Chang medium. G-banding technique was performed after cell fixation, and 450-band resolution was acquired. For every sample, 20 cells during metaphase were counted, which were derived from at least 2 isolated colonies.

### 2.3. DNA Extraction

DNA extractions on cord blood, amniotic fluid and placental tissue were performed for studies by array CGH, a short tandem repeat analysis and methylation-specific multiplex ligation-dependent probe amplification (MS-MLPA). QIAamp DNA mini kit was used according to the manufacturers’ guide.

### 2.4. Array Comparative Genomic Hybridization (Array CGH)

We used the SurePrint G3 ISCA V2 Human CGH 8X60K Array Kit (Agilent Technologies, Santa Clara, CA, USA) to identify genomic coping number variation (CNV) in testing sample. After DNA extraction, fluorescent labeling was performed with a SureTag DNA Labeling Kit (Agilent Technology). Cy5-dUTP was applied on the examined tissues, and Cy3-dUTP was applied on reference human genomic DNA (Male-Promega G147A, Female-Promega G152A). After hybridization with microarray chip, the result was read via a SureScan Microarray scanner (Agilent Technology) and analyzed with Agilent CytoGenomics, edition 2.7.8.0. A mosaic mode analyzer was used on dysplastic and non-dysplastic placenta due to known androgenetic/biparental mosaicism. The ratio of two mosaic cell lines was calculated using mean log ratio by experienced technologist (Lee, Chien-Hong).

### 2.5. Histology

Between 1 and 4 representative sections were selected from the dysplastic and the non-dysplastic part of the placenta. After formalin-fixed and paraffin-embedded tissue section, P57 immunohistochemistry was performed using Anti-p53 Antibody (clone DO-7, Leica biosystems, Buffalo Grove, IL, USA). The nuclear staining of stromal cell was reviewed by two pathologists.

### 2.6. Short Tandem Repeat (STR) Analysis

We used GlobalFiler Express PCR Amplification kit (Thermo Fisher, Waltham, MA, USA), GenePhile X-Plex PCR Amplification Kit (GenePhile Bioscience Co., Ltd., Taipei City, Taiwan) and 3500 DX Genetic Analyzer (Thermo Fisher) to count different tetranucleotide repeats and distinguish maternal or paternal origin of the examined alleles. In autosomal chromosomes, GlobalFiler Express PCR Amplification kit (Thermo Fisher) designs 21 STR loci using (D3S1358, vWA, D16S539, CSF1PO, TPOX, D8S1179, D21S11, D18S51, D2S441, D19S433, TH01, FGA, D22S1045, D5S818, D13S317, D7S820, SE33, D10S1248, D1S1656, D12S391, D2S1338) as markers, while 13 loci (DXS6807, DXS8378, DSX9902, DXS7132, DXS9898, DXS6809, DXS6789, DXS7424, DXS101, GATA172D05, HPRTB, DXS8377, DXS7423) on X chromosome were applied by GenePhile X-Plex PCR Amplification Kit (GenePhile Bioscience Co., Ltd.) [5]. The data were managed with Genescan (software), and the biparental-to-androgenetic ratio was calculated using the under-curve area by experienced technologist (Lee, Chien-Hong).

### 2.7. Methylation-Specific Multiplex Ligation-Dependent Probe Amplification (MS-MLPA)

To enable a diagnosis of the Beckwith–Weidemann syndrome, MS-MLPA was performed with SALSA MS-MLPA Probemix specific kit ME030-C3 BWS/RSS to identify abnormal epigenetic methylation or coping number variation (CNV) in chromosome 11p15 BWS/RSS region, which contains multiple probes targeting two imprinted domains, IC1(H19DMR) and IC2(KvDMR), and two related genes, the H19 gene and the KCNQ1OT1 gene.

### 2.8. Mosaic Analysis

Mean log ratio from array CGH and area under curve (AUC) from STR were used for mosaic analysis. In array CGH, assuming the mosaic phenomenon consisted of two genotypes, 46, X, i(X) (q10) and 45, X, we set δ for the portion of 46, X, i(X) (q10) and N for ploidy of chromosome Xq. The formula was designed as
(1)$$mean\;log\;ratio=-\log_{2}\frac{N}{2}\;$$
(2)N=3δ+(1−δ)=2δ+1
(3)$$\delta =\frac{N-1}{2}\times 100\%$$

In STR analysis, markers of autosomal chromosomes are used to evaluate biparental/androgenetic mosaicism, and markers of X chromosome are applied to estimate 46, X, isochromosome Xq/45, X mosaicism. By using the area under curve (AUC), the ratio of parental signals could be transferred to mosaic rate. From markers of autosomal chromosomes, taking P for parental signals, M for maternal signals and the portion of biparental cells (x) could be calculated as follows:(4)$$\frac{P}{M}=\frac{x+2\left(1-x\right)}{x}$$
(5)$$x=\frac{2M}{P+M}\times 100\%$$

From markers of X chromosome, taking P for parental signals, M for maternal signals and y as the portion of 46, X, i(X) (q10), the AUC ratio could be presented as
(6)$$\frac{P}{M}=\frac{y+\left(1-y\right)}{2y}$$
(7)$$y=\frac{M}{2P}\times 100\%$$

## 3. Results

After delivery, the placenta grossly presented with two kinds of morphology: a dysplastic part with grape-like, multicystic change, and a non-dysplastic part with near-normal appearance. The ratio of the area over the dysplastic and the non-dysplastic part was about 7:1, and the umbilical cord was centrally inserted and divided symmetrically to the whole placenta (Figure 2). Under the hematoxylin–eosin (H and E) stain, the dysplastic placenta showed abnormally enlarged villi with cavernous cysts and loose myxoid stroma (Figure 3a). Abnormal chorangiomatosis-like vascular proliferation within some villi was seen (Figure 3b). There was no significant trophoblastic proliferation, which was compatible with the diagnosis of placenta mesenchymal dysplasia rather than molar pregnancy. Immunohistochemistry stain for p57 showed a complete loss of expression of the stromal cells in the dysplastic part of the placenta, indicating a loss of expression from maternal genome (Figure 3c,d) [6].

Karyotype of cord blood, non-dysplastic placenta and dysplastic placenta showed diverse results. Cord blood was karyotyped as 46, X, i(X) (q10), but non-dysplastic and dysplastic placenta showed 45, X without mosaicism. These two kinds of genotype were assumed to be the main composition of the gestational products, including the placenta, amniotic fluid and fetus, in the following analysis.

An array CGH of uncultured cells from cord blood, non-dysplastic and dysplastic placenta revealed inconsistent copy number variation of X chromosome. By calculation, portions of 46, X, i(X) (q10) in cord blood, non-dysplastic and dysplastic placenta were 98.69%, 69.01%, 22.43%, and portions of 45, X were 1.31%, 30.99%, 77.57%, respectively. Amniotic fluid from prenatal amniocentesis showed totally with 46, X, i(X) (q10) (Table 1).

The analysis of short tandem repeats (STR) helps to distinguish parental origin of the genome, as shown in Figure 4. Markers of autosomal chromosomes are located in chromosomes 1 to 21, and markers of sex chromosomes are located on both the p arm and q arm of X chromosome. Through a comparison with parents’ STR number, the result showed that all products of conception shared the same allelic composition, indicating the single zygotic origin. However, the different ratio of paternal and maternal signals from specimens implied possible androgenetic/biparental mosaicism. Particularly in the dysplastic placenta, paternal signals were significantly greater than maternal signals, suggesting an androgenetic-dominant composition. In Table 2, STR markers on autosomes were using to demonstrate androgenetic/biparental mosaic ratio by calculating the area under curve (AUC). The cord blood, amniotic fluid and non-dysplastic placenta showed nearly 100% biparental cells. However, the portion of biparental cells in the dysplastic placenta was only 25.80%.

In addition, the absence of markers on the p arm of maternal X chromosome reveals the maternal origin of isochromosome Xq (Table 3). The proportion of isochromosome Xq in the non-dysplastic placenta and the dysplastic placenta were 65.25% and 8.16%, respectively. On the other hand, the amniotic fluid and cord blood showed nearly totally 46, X, isochromosome Xq. The result was compatible with array CGH (Table 1).

Finally, MS-MLPA of cord blood was performed and showed no abnormal finding. Beckwith–Wiedemann syndrome was then excluded due to normal fetal growth and normal methylation of the BWS/RSS region (Figure 5).

## 4. Discussion

The incidence of PMD has been estimated as ranging from 0.002% [7] to 0.02% [7,8], and the incidence of genotype 46, X, i(Xq) has been reported as 1 in 25,000 live baby births [9]. It is extremely rare to have these two situations happening coincidentally. From a systemic review [2], 28% of cases with PMD have abnormal genetic evaluation of the fetus, including chromosomal anomaly, Beckwith–Wiedemann syndrome or uniparental disomy. Chromosomal anomalies have been reported with trisomy 13, Klinefelter’s syndrome (47, XXY), 69, XXX and mosaicism for trisomy 13. To our knowledge, this is the first case presented with PMD with isochromosome Xq. The STR analysis of X chromosome shows double dosage of single maternal Xq signals, which hints at the maternal origin of isochromosome Xq and indicates malsegregation, and the formation of isochromosome occurs at the moment of fertilization, during which meiosis 2 of oocytogenesis completes.

Additionally, this is also the first reported case presenting placental mesenchymal dysplasia with a karyotype of 45, X in the dysplastic placenta. Only one case has been diagnosed where a complete hydatidiform mole displayed a 45, X karyotype [10]. In our case, a distinctly decreased dosage on the whole X chromosome from array CGH on uncultured dysplastic and non-dysplastic placenta, indicating a larger portion of cells with 45, X. A karyotype of the dysplastic placenta further confirmed the result. By using STR markers, only 25.8% cells of dysplastic placenta showed biparental genome (Table 2), and 74.2% cells presented paternal alleles only. On the other hand, STR markers from X chromosome revealed 65.25% and 8.16% cells of non-dysplastic and dysplastic placenta, having isochromosome Xq, which means about 34.75% and 91.84% cells are karyotyped as 45, X in these two parts of the placenta, respectively. Because of the merely 25% of biparental alleles seen in the dysplastic placenta and the different composition of biparental/androgenetic mosaicism and 46, X, isochromosome Xq/45, X mosaicism, we believe most cells in the dysplastic placenta should be a uniparental 45, X (paternal), which may come from a sperm with haploid endoduplication and subsequent failure of replication of X chromosome (Figure 6).

It is worth mentioning that the STR analysis showed uniform paternal and maternal signals from somatic chromosome to sex chromosome, suggesting the zygote carries one haploid from a single sperm and the other one from a single oocyte. Under such conditions, the diversity of different genotypes in different gestational tissues should be described as “mosaicism” instead of “chimerism”. Androgenetic/biparental mosaicism causing placental mesenchymal dysplasia was first announced by K. A. Kaiser-Rogers in 2006 [3]. In that study, microsatellite markers for chromosome 7, 9, 12, 14, 16, X were applied to check the parental origin of the genome from amniotic fluid, fetus and chorions, and they found an increased proportion of androgenetic cells in chorionic mesenchyme and enlarged chorionic vessels on the placenta, from 80% to 100% [3]. In our case, the androgenetic rate is also higher in the dysplastic placenta (74.20%, androgenetic rate = 1 − biparental rate, Table 2), and nearly no androgenetic cells are seen in the fetal cord blood by using STR analysis.

According to Aspasia Destouni, et al. [11], several post-zygotic events contribute to chromosomal instability and may lead to mosaicism or mixoploidy. Androgenetic mixoploidy, which is defined as cells with only paternally derived genome and different ploidy status, can be seen in human diseases, including PMD, complete hydatidiform mole or partial hydatidiform moles. In their study, they named the phenomenon of spontaneous segregation of whole paternal genomes from biparental zygote into distinct cell linages as “heterogoneic cell division”, which was proven in the bovine exam. That may be a reasonable mechanism in our case as well, explaining why androgenetic 45, X forms. With regard to the mosaicism of biparental 46, X, isoXq/45X, it would be due to an oocyte carrying only the q arm on X chromosome and a failed centromere division during metaphase II, according to David F. Callen’s proposed mechanism [12].

The limitations of these genetic evaluations are the following: (1) During cytogenetic cultures, some cells with uniparental genome or aneuploidy may be hard to grow, thus we cannot identify the definite genotypes of all cell lines; (2) Microscopically, there are still a lot of normal villi around the dysplastic cysts, so the sample is mixed and contaminated. If available, microdissection should be considered to isolate the truly dysplastic cells; (3) In our study, only cord blood and amniotic fluid of the fetus were taken for evaluation, which is not sufficiently representative of all germ layers. In addition, only one specimen on each site of the placenta was collected, thus the within-group variation could not be addressed.

## 5. Conclusions

This is a unique case, which presented PMD combined with mosaic biparental 46, X, iso(Xq) and androgenetic 45, X. The highest ratio of cells with androgenetic (45, X) was detected in the dysplastic placenta.

## Figures and Tables

**Figure 1 genes-13-00245-f001:**
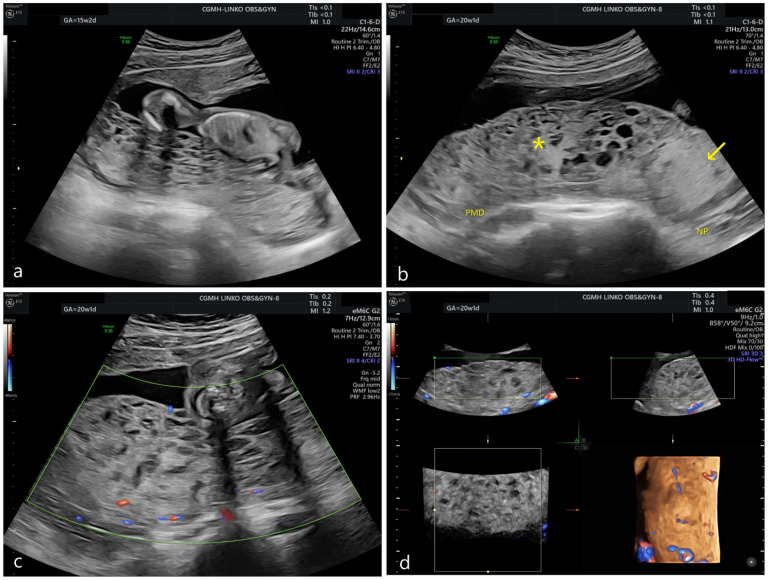
Prenatal ultrasonography. (**a**) At 15 weeks of gestation, multicystic change of placenta could be seen. (**b**) At 20 weeks of gestation, an enlarged dysplastic portion of the placenta was detected. (star *: dysplastic placenta; arrow: non-dysplastic placenta; PMD: placental mesenchymal dysplasia; NP: normal placenta) (**c**) Under Doppler scan, the dysplastic placenta showed low vascularity. (**d**) Under 3D reconstruction, the dysplastic placenta remained avascular.

**Figure 2 genes-13-00245-f002:**
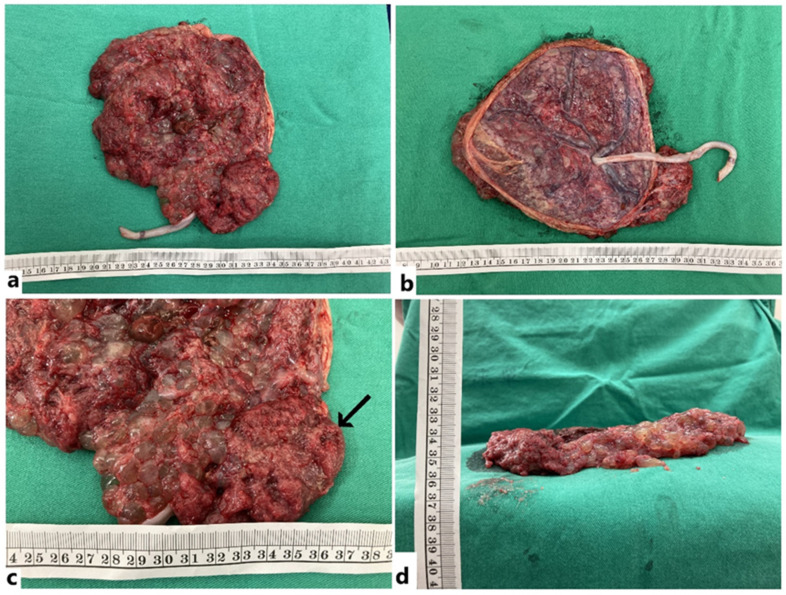
Gross appearance of the placenta and the fetus. (**a**) Maternal side; (**b**) Fetal side with centrally inserted umbilical cord; (**c**) Small portion of non-dysplastic placenta (arrow); (**d**) Lateral view of placenta.

**Figure 3 genes-13-00245-f003:**
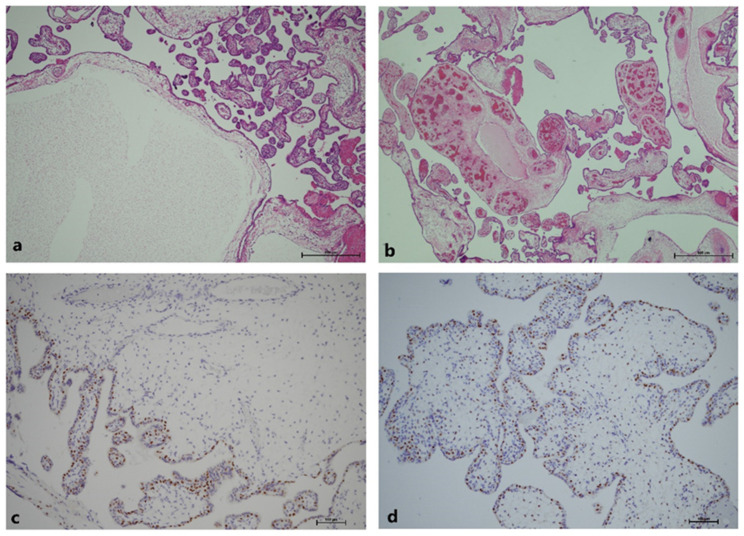
Histology of dysplastic placenta. (**a**) Dysplastic villi under hematoxylin–eosin stain. An edematous, dysplastic villi was surrounded by some normal villi, showing a mosaic pattern of histological appearance; (**b**) Chorangiosis in dysplastic villi; (**c**) P57 stain of dysplastic villi. No nuclear uptake in stroma is seen, indicating no expression of maternal genome; (**d**) P57 stain of normal villi. Nuclear stain in stroma is seen.

**Figure 4 genes-13-00245-f004:**
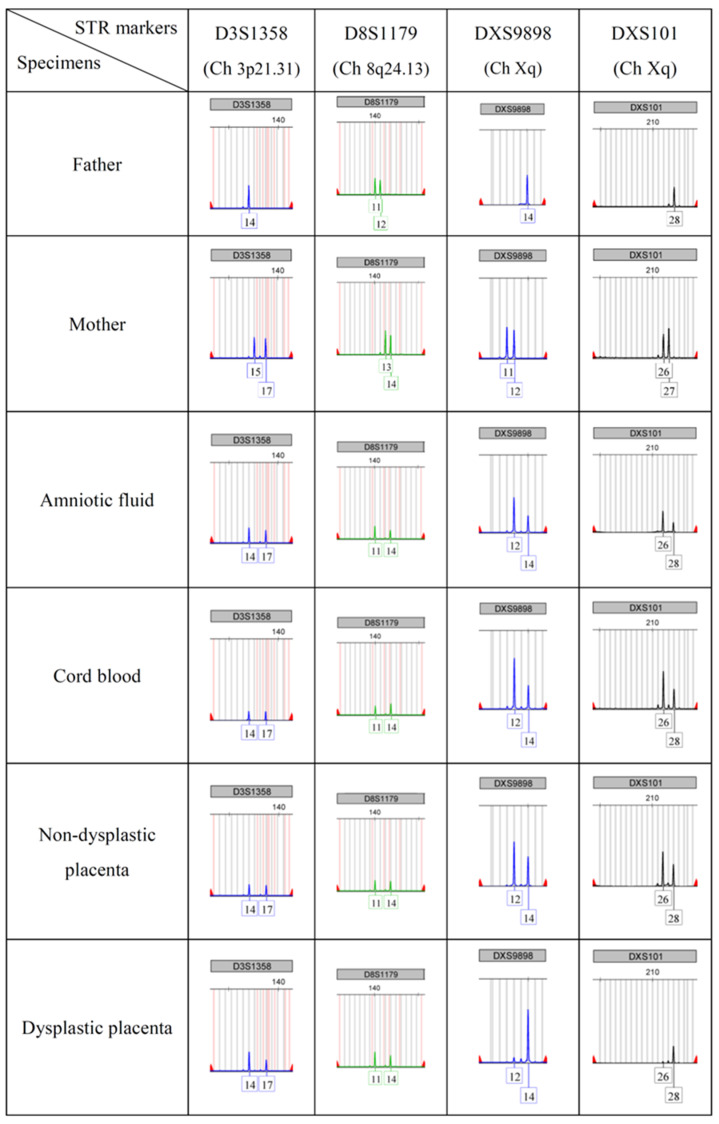
Short tandem repeat (STR) of blood of father, blood of mother, amniotic fluid, cord blood, non-dysplastic placenta and dysplastic placenta. All products of conception shared the same allelic signal.

**Figure 5 genes-13-00245-f005:**
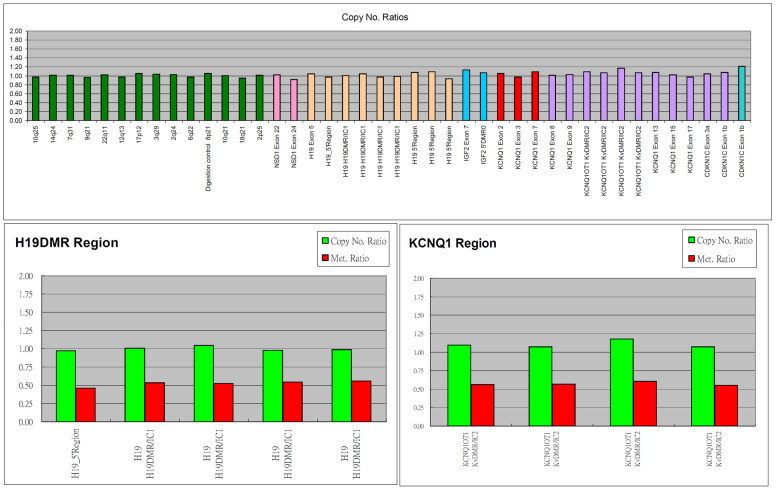
Multiplex ligation-dependent probe amplification (MLPA) of amniotic fluid showed normal methylation of BWS/RSS region.

**Figure 6 genes-13-00245-f006:**
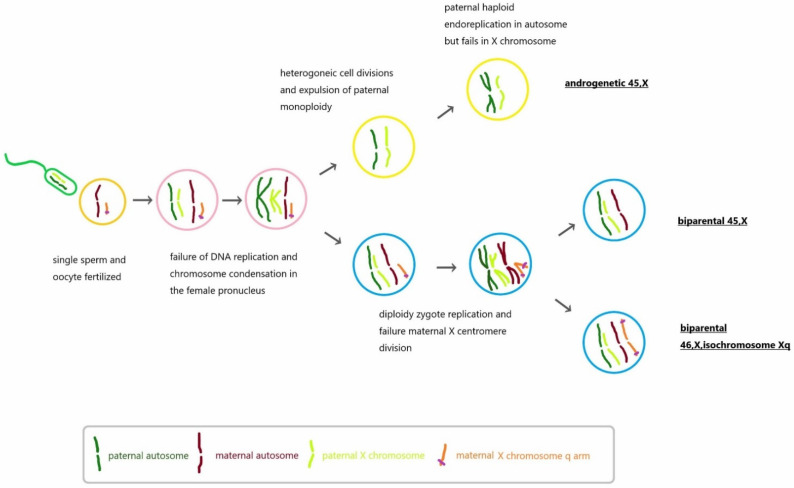
Hypothetical cause of biparental/androgenetic mosaicism and 46, X, isochromosome Xq/45, X mosaicism. Single sperm was fertilized with the oocyte carrying 22, Xq. Before the female pronucleus condensed, the paternal genome replicated first and was expelled as distinct cell, which is called heterogeneous cell division. The autosomes of paternal haploid endoreplicated but failed in X chromosome, resulting in the androgenetic 45, X. On the other hand, after heterogeneous cell division, the biparental zygote went through cell replication and mis-division of the centromere on X chromosome, leading to biparental 46, X, isoXq/45, X mosaicism.

**Table 1 genes-13-00245-t001:** Mosaic analysis of array CGH.

	Amniotic Fluid	Cord Blood	Non-Dysplastic Placenta	Dysplastic Placenta
Mean log ratio	0.591397	0.572296	0.251074	−0.46537
N/2	1.506705	1.486888	1.190093	0.724287
N	3.01341	2.973776	2.380186	1.448574
δ	100.67%	98.69%	69.01%	22.43%
1−δ	−0.67%	1.31%	30.99%	77.57%

Mean log ratio: −log2 (N/2). N: ploidy of chromosome Xq. δ: the portion of 46, X, i(X) (q10). 1−δ: the portion of 45, X.

**Table 2 genes-13-00245-t002:** Mosaic analysis of androgenetic/biparental cell lines using STR markers from autosomal chromosomes.

		Amniotic Fluid			Cord Blood			Non-Dysplastic Placenta			Dysplastic Placenta		
STR Markers	Chromosomal Location	Allele (P/M)	AUC (P/M)	Biparental Rate	Allele (P/M)	AUC (P/M)	Biparental Rate	Allele (P/M)	AUC (P/M)	Biparental Rate	Allele (P/M)	AUC (P/M)	Biparental Rate
D3S1358	3p21.31	14/17	16,751/14,151	91.59%	14/17	18,393/17,785	98.32%	14/17	15,856/14,804	96.57%	14/17	48,076/6164	22.73%
vWA	12p13.31	17/18	27,518/24,229	93.64%	17/18	41,960/33,329	88.54%	17/18	27,448/30,097	104.60%	17/18	66,220/11,697	30.02%
D16S539	16q24.1	9/11	32,759/36,536	105.45%	9/11	78,624/76,666	98.74%	9/11	66,014/61,394	96.37%	9/11	120,006/11,389	17.34%
D8S1179	8q24.13	11/14	16,840/11,381	80.66%	11/14	18,445/22,999	110.99%	11/14	18,056/16,524	95.57%	11/14	48,812/6321	22.93%
D18S51	21q11.2–q21	15/16	93,826/104,530	105.40%	15/16	149,449/118,218	88.33%	15/16	102,685/98,713	98.03%	15/16	158,985/31,591	33.15%
D2S441	18q21.33	11/13	39,236/27,800	82.94%	11/13	47,803/47,727	99.92%	11/13	41,749/43,326	101.85%	11/13	86,278/14,227	28.31%
TH01	2p14	9/7	42,235/53,801	112.04%	9/7	70,117/80,395	106.83%	9/7	63,412/66,071	102.05%	9/7	109,520/18,859	29.38%
FGA	19q12	20/23	45,652/55,661	109.88%	20/23	97,451/96,252	99.38%	20/23	69,714/83,480	108.99%	20/23	111,815/14,095	22.39%
D5S818	11p15.5	9/13	36,069/24,468	80.84%	9/13	38,269/37,060	98.40%	9/13	25,995/25,416	98.87%	9/13	78,297/10,763	24.17%
D13S317	4q28	10/11	42,997/42,240	99.11%	10/11	70,928/54,565	86.96%	10/11	37,985/39,182	101.55%	10/11	95,033/13,547	24.95%
SE33	5q21–31	27.2/18	128,927/130,009	100.42%	27.2/18	213,111/210,357	99.35%	27.2/18	180,405/160,701	94.22%	27.2/18	218,007/48,467	36.38%
D10S1248	13q22–31	14/15	18,771/14,955	88.69%	14/15	18,719/19,616	102.34%	14/15	13,096/11,450	93.29%	14/15	49,175/3700	14.00%
D12S391	6q14	23/18	40,624/42,025	101.70%	23/18	55,074/72,839	113.89%	23/18	37,519/42,589	106.33%	23/18	98,544/16,879	29.25%
D2S1338	10q26.3	19/24	67,167/65,573	98.80%	19/24	69,352/61,456	93.96%	19/24	63,880/65,648	101.36%	19/24	159,569/23,995	26.14%
		Average		96.51%	Average		99.00%	Average		99.98%	Average		25.80%

Assume mosaicism with (46, X, i(X) (q10), biparental) and (45, X, androgenetic). biparental rate = 2M/(P + M).

**Table 3 genes-13-00245-t003:** Mosaic analysis of 46, X, iso(Xq) and 45, X using STR markers from sex chromosomes.

		Amniotic Fluid			Cord Blood			Non-Dysplastic Placenta			Dysplastic Placenta		
STR Markers	Chromo-Somal Location	Allele (P/M)	AUC (P/M)	46, X, isoXq Rate	Allele (P/M)	AUC (P/M)	46, X, isoXq Rate	Allele (P/M)	AUC (P/M)	46, X, isoXq Rate	Allele (P/M)	AUC (P/M)	46, X, isoXq Rate
DXS6807	Xp	11/x(15)	12,848/0		11/x(15)	55,158/0		11/x(15)	49,061/0		11/x(15)	97,868/0	
DXS9902	Xp	10/x(11)	20,816/0		10/x(11)	47,862/0		10/x(11)	51,007/0		10/x(11)	83,574/0	
DXS9898	Xq	14/12	15,843/32,626	102.97%	14/12	60,309/126,386	104.78%	14/12	54,856/81,814	74.57%	14/12	112,201/11,273	5.02%
DXS8377	Xq	48/51	25,374/49,396	97.34%	48/51	65,812/134,888	102.48%	48/51	68,455/83,375	60.90%	48/51	142,370/17,128	6.02%
HPRTB	Xq	15/12	28,132/53,429	94.96%	15/12	62,571/149,412	119.39%	15/12	71,063/100,667	70.83%	15/12	145,559/23,456	8.06%
DXS7132	Xq	14/15	36,437/60,358	82.83%	14/15	119,612/162,804	68.06%	14/15	88,222/112,703	63.87%	14/15	165,957/26,683	8.04%
DXS101	Xq	28/26	17,510/37,195	106.21%	28/26	47,461/92,150	97.08%	28/26	41,584/65,929	79.27%	28/26	93,974/20,712	11.02%
DXS6789	Xq	15/21	69,262/142,895	103.16%	15/21	175,639/22,9088	65.22%	15/21	164,974/190,858	57.84%	15/21	245,485/43,987	8.96%
DXS7424	Xq	15/16	18,594/25,692	69.09%	15/16	38,859/61,909	79.66%	15/16	43,348/42,893	49.48%	15/16	66,848/13,377	10.01%
		Average		93.79%	Average		90.95%	Average		65.25%	Average		8.16%

Assume mosaicism with (46, X, i(X) (q10)) and (45, X). 46, X, isoXq rate = M/2P.

## Data Availability

The datasets obtained and/or analyzed during the current study are available from the corresponding author on reasonable request.

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
