# Peer review of "Prenatal Diagnosis of Placental Mesenchymal Dysplasia with 46, X, Isochromosome Xq/45, X Mosaicism"

_genes, 2022, doi:10.3390/genes13020245_

Round 1
Reviewer 1 Report
#1. The authors divided the villous tissue into two regions, non-dysplastic and dysplastic villi, based on gross findings. In contrast to non-dysplastic villi, stromal p57 was absent in dysplastic villi, and paternal disomy-dominant mosaicism was indicated.
The molecular genetic essence of PMD is a mosaicism of paternal disomy.
PMD is both pathologically and genetically heterogeneous, and the distribution of grossly or histologically dysplastic areas does not always correspond to the paternal disomy-riched area. Though 1 to 4 representative sections were collected from dysplastic part and non-dysplastic part of the villi for histological study, but it did not mention how many samples were used for molecular genetic studies.
It is necessary to indicate how many specimens were collected from each of the non-dysplastic and dysplastic villi for DNA extraction, and how much the mosaic ratio varied among the specimens. If only one specimen was collected from each site, this should be shown as a limitation of the study.
#2. The authors mentioned that this was the first reported case presenting PMD with a karyotype of 45,X. Although it may be interesting, it was not discussed whether the X monosomy had affected on the development of PMD, or just a coincidence. The authors also stated that it was extremely rare for a 46,X,i(Xq) fetus in PMD cases, and gave the probability as the number obtained by multiplying the PMD by the probability of 46,X,i(Xq)at birth, however, most PMDs without Beckwith-Wiedemann syndrome are related to the female sex of the child, so it should not be a simple multiplication.
#3. Though the Figure 7 is interesting because it shows the molecular genetic origin of each component in an easy-to-understand manner, this figure may mislead people into the idea that androgenic cells differentiate into trophoblast cells and biparental cells differentiate into embryo. Isn't it true that both androgenic and biparental cells start to differentiate into embryos, and only amphipathic cells remained (rescued)?
Author Response
#1. The authors divided the villous tissue into two regions, non-dysplastic and dysplastic villi, based on gross findings. In contrast to non-dysplastic villi, stromal p57 was absent in dysplastic villi, and paternal disomy-dominant mosaicism was indicated.
The molecular genetic essence of PMD is a mosaicism of paternal disomy.
PMD is both pathologically and genetically heterogeneous, and the distribution of grossly or histologically dysplastic areas does not always correspond to the paternal disomy-riched area. Though 1 to 4 representative sections were collected from dysplastic part and non-dysplastic part of the villi for histological study, but it did not mention how many samples were used for molecular genetic studies.
It is necessary to indicate how many specimens were collected from each of the non-dysplastic and dysplastic villi for DNA extraction, and how much the mosaic ratio varied among the specimens. If only one specimen was collected from each site, this should be shown as a limitation of the study.
Answer: Thank for the very important comment. We took only one sample from the dysplastic and nondysplastic area of placenta as judged from their outer morphology. The limitation has been addressed in the last paragraph of discussion, page 11. If we have next case, we will try the very best to take more than 3 specimen from every parts of the placenta and find the variation.
#2. The authors mentioned that this was the first reported case presenting PMD with a karyotype of 45,X. Although it may be interesting, it was not discussed whether the X monosomy had affected on the development of PMD, or just a coincidence. The authors also stated that it was extremely rare for a 46,X,i(Xq) fetus in PMD cases, and gave the probability as the number obtained by multiplying the PMD by the probability of 46,X,i(Xq)at birth, however, most PMDs without Beckwith-Wiedemann syndrome are related to the female sex of the child, so it should not be a simple multiplication.
Answer: Thank you for pointing out where it may be inappropriate. Indeed, the incidence of PMD combinined with 45,isoXq is extremely rare, and no definite mechanism has been documented so far. It’s too rough to multiply two incidences of the two diagnoses, but this case is exactly the first reported PMD combined with 46,X,isoXq. We’ve adjusted the description in the first paragraph of discussion, page 9.
#3. Though the Figure 7 is interesting because it shows the molecular genetic origin of each component in an easy-to-understand manner, this figure may mislead people into the idea that androgenic cells differentiate into trophoblast cells and biparental cells differentiate into embryo. Isn't it true that both androgenic and biparental cells start to differentiate into embryos, and only amphipathic cells remained (rescued)?
Answer: It’s true that the figure misleads readers to think in a bit confusing direction. We decided to take off the figure 7, and adjusted the figure 6 to be more comprehensive. Another paragraph was added in discussion, page 10-11, which quoted the suggesting reference from the other reviewer. We believed that both androgenic and biparental cells persist in whole gestational tissue, including the placenta and the fetus. Somehow the baby portion has more normal biparental cells, which may prove beneficial for adequate function for embryonic proliferation.
Reviewer 2 Report
The date present by Hsu et al. contains a description of an interesting and rare case of the placental mesenchymal dysplasia with androgenetic/biparental mosaicism. The methods used and the description of the results are adequate and confirm the conclusions of the study.
But the manuscript needs some changes.
- In Figure 6, it is necessary to present in more detail all the stages of segregation errors, in particular the loss of the X chromosome of paternal and maternal origin.
- To enhance the discussion of their results, the authors are encouraged to use the results of fundamental research such as article of Destouni A. et al. Zygotes segregate entire parental genomes in distinct blastomere lineages causing cleavage-stage chimerism and mixoploidy //Genome Res. 2016 May; 26(5): 567–578.
Author Response
The date present by Hsu et al. contains a description of an interesting and rare case of the placental mesenchymal dysplasia with androgenetic/biparental mosaicism. The methods used and the description of the results are adequate and confirm the conclusions of the study.
But the manuscript needs some changes.
- In Figure 6, it is necessary to present in more detail all the stages of segregation errors, in particular the loss of the X chromosome of paternal and maternal origin.
Answer: Thank for the professional suggestion! We tried to come up with the possible mechanisms of androgenic 45,X and biparental 45,X, and found 2 references to describe the proposed process. Figure 6 has been adjusted based on the reported hypothesis, and it’s indeed very hard to explain how androgenic 45,X is formed.
- To enhance the discussion of their results, the authors are encouraged to use the results of fundamental research such as article of Destouni A. et al. Zygotes segregate entire parental genomes in distinct blastomere lineages causing cleavage-stage chimerism and mixoploidy //Genome Res. 2016 May; 26(5): 567–578.
Answer: Thank for the very precious reference! We appreciate the biggest powerful explanation of the complex story! We have added this manuscript as the reference 12 for our manuscript.